# Paternal nicotine exposure alters hepatic xenobiotic metabolism in offspring

Markus P Vallaster[1], Shweta Kukreja[1], Xin Y Bing[1], Jennifer Ngolab[2,3], Rubing Zhao-Shea[2,3], Paul D Gardner[2,3], Andrew R Tapper[2]*, Oliver J Rando[1]*

[1]Department of Biochemistry and Molecular Pharmacology, University of Massachusetts Medical School, Worcester, United States; [2]Brudnick Neuropsychiatric Research Institute, Department of Psychiatry, University of Massachusetts Medical School, Worcester, United States; [3]Department of Psychiatry, University of Massachusetts Medical School, Worcester, United States

**Abstract** Paternal environmental conditions can influence phenotypes in future generations, but it is unclear whether offspring phenotypes represent specific responses to particular aspects of the paternal exposure history, or a generic response to paternal 'quality of life'. Here, we establish a paternal effect model based on nicotine exposure in mice, enabling pharmacological interrogation of the specificity of the offspring response. Paternal exposure to nicotine prior to reproduction induced a broad protective response to multiple xenobiotics in male offspring. This effect manifested as increased survival following injection of toxic levels of either nicotine or cocaine, accompanied by hepatic upregulation of xenobiotic processing genes, and enhanced drug clearance. Surprisingly, this protective effect could also be induced by a nicotinic receptor antagonist, suggesting that xenobiotic exposure, rather than nicotinic receptor signaling, is responsible for programming offspring drug resistance. Thus, paternal drug exposure induces a protective phenotype in offspring by enhancing metabolic tolerance to xenobiotics.

*For correspondence: Andrew. Tapper@umassmed.edu (ART); Oliver.Rando@umassmed.edu (OJR)

**Competing interests:** The authors declare that no competing interests exist.

## Introduction

Environmental conditions experienced in one generation can affect phenotypes that manifest in future generations, a phenomenon sometimes referred to as the 'inheritance of acquired characters.' In mammals, a substantial body of literature links various maternal exposures to offspring phenotypes (*Harris and Seckl, 2011*; *Rando and Simmons, 2015*; *Simmons, 2011*), and an increasing number of studies have shown that *paternal* environment can also alter offspring phenotype (*Rando, 2012*). Paternal effect paradigms are of particular mechanistic interest in mammals, given that it is challenging to disentangle maternal environment effects on the oocyte epigenome from effects on uterine provisioning during offspring development. In contrast, in many paternal effect paradigms, males contribute little more than sperm to the offspring, simplifying the search for the mechanistic underpinnings of paternal effects on children. A large number of paternal exposure paradigms have been used to show that a father's diet can affect metabolic phenotypes in the next generation (*McPherson et al., 2014*; *Rando, 2012*), while another large group of studies link paternal stress (using paradigms such as social defeat stress, or early maternal separation) to anxiety-related behaviors and cortisol release in offspring (*Bale, 2015*). Finally, a growing number of toxins and drugs have been shown to induce effects on various offspring phenotypes (*Skinner et al., 2011*; *Vassoler et al., 2013*; *Yohn et al., 2015*; *Zeybel et al., 2012*).

A key challenge in such studies at present is to understand how the offspring phenotype is related to the stimulus presented in the paternal generation – in other words, how specific is the offspring response? This challenge is compounded by the fact that many of the stimuli used for

**eLife digest** Until recently, it seemed impossible that the conditions a person or animal experiences during their lifetime might affect the health of their offspring and future generations. Research over the past decade, however, has shown that a parent's environment can cause changes that can be passed to future generations. For example, studies in rodents have shown that a father's diet influences the way their offspring metabolize food. Moreover, a male mouse exposed to stress or toxins fathers pups that often respond differently in stressful situations relative to other mice.

So, how do these traits get transferred to offspring via sperm and how specific is the next generation's response to the environmental pressures faced by their fathers? Many studies so far have looked at environmental influences that may have broad biological effects, for example a high fat diet. Now, some scientists are trying to understand whether exposure to nicotine, which has a more targeted effect, causes drug-specific effects in offspring.

Vallaster et al. now show that mice whose fathers had been exposed to nicotine before mating are more able to withstand toxic levels of the chemical than mice whose fathers were never exposed to the drug. In the experiments, some male mice were given water with nicotine in it over the course of five weeks. Later, the offspring of these mice were exposed to nicotine to see whether they were more or less sensitive to it than offspring of unexposed males. It turns out the mice with nicotine-exposed fathers have a higher resistance to the toxic effects of nicotine and, unexpectedly, to toxic levels of cocaine as well. This suggests that the pups of nicotine-exposed fathers are not specifically programmed to respond to nicotine, but instead are more resistant to toxins in general.

Vallaster et al. found that the livers of the offspring of nicotine-exposed fathers appear to be better able to metabolize both drugs. Exposing the fathers to another drug called mecamylamine (which can prevent many of nicotine's effects on the body) also made their offspring more resistant to nicotine, showing that multiple drugs may make offspring more toxin-resistant. Studies in humans will be needed to confirm whether a father's nicotine use affects children the same way it does mice. Similar mice studies also may help scientists to study how other types of environmental exposure might affect a man's future children.

paternal effect paradigms – low protein and high fat diets, social stressors, and endocrine disruptors – have pleiotropic effects on organismal physiology. We therefore sought to develop a paternal effect paradigm based on a defined ligand-receptor interaction, to enable pharmacological interrogation of the specificity of the offspring phenotype. Nicotine is a commonly-used drug in humans, and acts by binding to and activating nicotinic acetylcholine receptors (nAChRs), ligand-gated cation channels normally activated by the endogenous neurotransmitter acetylcholine. Maternal use of nicotine has been linked to multiple phenotypes in offspring (*Yohn et al., 2015*; *Zhu et al., 2014*), and although effects of paternal nicotine exposure have been less-studied, paternal smoking in humans has been suggested to affect metabolic phenotypes in children (*Pembrey et al., 2006*).

Here, we develop a rodent model for paternal nicotine effects, asking (1) whether exposure of male mice to nicotine could impact phenotypes in offspring, and (2) whether any affected phenotype would be specific for nicotine. We found that paternal exposure to nicotine induced a protective response in the next generation, as male offspring of nicotine-exposed fathers exhibited significant protection from nicotine toxicity. Importantly, this toxin resistance was not specific to nicotine, instead reflecting a more general xenobiotic response – offspring of nicotine-exposed fathers exhibited increased hepatic expression of a variety of genes involved in clearance of xenobiotics, and these animals were resistant to cocaine as well as to nicotine toxicity. Finally, we found that enhanced resistance to nicotine toxicity was also observed in offspring of males treated with the nicotine antagonist mecamylamine, strongly suggesting that drug resistance in offspring is a common outcome of paternal exposure to multiple xenobiotics rather than a specific response arising from nicotine signaling. Taken together, our results describe a novel paternal effect paradigm, and demonstrate that in the case of paternal nicotine exposure, the phenotype observed in offspring is a relatively generic response – enhanced xenobiotic resistance – rather than a selective downregulation of the specific molecular pathway subject to paternal perturbation.

## Results

### Effects of paternal exposure history on offspring nicotine sensitivity

We established a paternal exposure paradigm in which male mice were either provided with nicotine hydrogen tartrate (nicotine 200 µg/ml free base, sweetened with saccharine) in their drinking water, or a control solution of tartaric acid and saccharine. Mice consumed nicotine or control solutions (NIC or TA, respectively) from 3 weeks of age until 8 weeks of age. As previously described (*Zhao-Shea et al., 2015*), this administration regimen maintains a high level of nicotine in the bloodstream (*Figure 1—figure supplement 1A–B*), and results in nicotine dependence in exposed animals (*Zhao-Shea et al., 2013*). Males were then withdrawn from nicotine for one week prior to mating in order to prevent any potential for seminal fluid transmission of nicotine (the half-life of nicotine in mice is ~10 min, the half-life of its 'long-lived' metabolite cotinine is ~40 min [*Siu and Tyndale, 2007*]). Nicotine and control males were then mated with control females. Overall, we observed no difference in average size or sex ratio of litters arising from control or nicotine matings, or in offspring body weights (*Figure 1—figure supplement 1C–F*).

We first sought to determine whether the enforced nicotine withdrawal in our exposure paradigm might result in a paternal stress response that could affect the phenotype of progeny. As anxiety-related behaviors have been reported in offspring of males subject to several distinct stress paradigms (*Dietz et al., 2011*; *Gapp et al., 2014*; *Short et al., 2016*) (albeit not all such paradigms – [*Rodgers et al., 2013*]), we therefore assessed anxiety behaviors in TA and NIC offspring. Importantly, we observed no differences between TA and NIC offspring in time spent in the center during an open field anxiety test, or in time spent or number of entries into the open arms of an elevated plus maze (*Figure 1—figure supplement 2*). These results and results discussed below (see Figure 6) indicate that our nicotine administration paradigm does not induce a stress response robustly enough, or for long enough prior to mating, to affect offspring phenotype.

We next asked whether paternal nicotine administration could more specifically affect nicotine-related phenotypes in the next generation. We first focused on a physiological readout of offspring sensitivity to nicotine, using a well-established assay for suppression of locomotor activity by acute nicotine administration (*Tapper et al., 2004*). Briefly, after acclimating animals to a saline injection protocol for three days, animals are injected with either nicotine (1.5 mg/kg) or saline, and immediately introduced to a novel environment. Saline-injected animals actively explore the novel environment, and locomotor activity is quantified over a 40 min time course (*Figure 1* – Baseline). In this paradigm, injection of nicotine results in rapid suppression of locomotor activity, followed by a gradual recovery of exploratory behavior over the time course of the assay. Using this assay, we observed no significant difference in nicotine sensitivity between TA and NIC offspring, either for male or female offspring (*Figure 1*, *Figure 1—figure supplement 3*). We therefore conclude that the acute locomotor suppression response to nicotine is not altered by our paternal nicotine exposure paradigm.

We next sought to identify any effects of paternal nicotine exposure on nicotine reinforcement in offspring using an operant self-administration assay (*Fowler et al., 2011*). Here, after surgical implantation of a catheter into the superior vena cava, animals are subject to caloric restriction and trained to nose-poke an active portal to self-administer (SA) sucrose. TA and NIC offspring exhibited similar behavior during the training period, with the exception of a modest albeit significant difference in sucrose SA on the final day of dietary training (*Figure 2—figure supplement 1*). After seven days of food shaping, animals were placed in the operant chamber, a nicotine infusion pump was connected to the central catheter, and the dietary reward for nose-poking the active portal was replaced with a nicotine infusion. The amount of nicotine self-administered every day was then measured per session over the course of 35 days, with the nicotine infusion dose increasing every 4–8 days (Materials and methods). Overall, there was no difference in daily nicotine SA between offspring of control males and offspring of nicotine-exposed males (*Figure 2A*), indicating that nicotine reward behavior is not significantly reprogrammed by our paternal exposure paradigm.

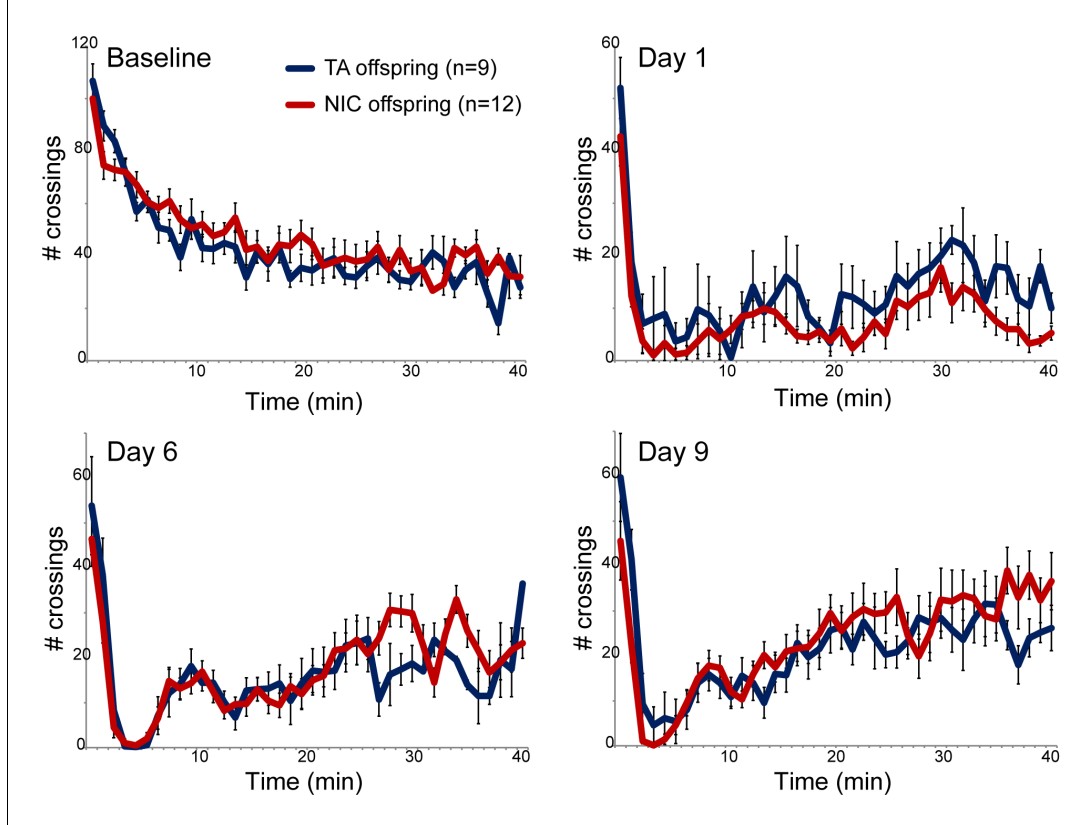

**Figure 1.** Nicotine suppression of locomotor activity is unaffected by paternal nicotine history. Nicotine effects on locomotor activity were assayed in male offspring of control (TA) or nicotine-exposed (NIC) fathers. Data for females and alternative administration regimens are shown in *Figure 1—figure supplement 3*. For each plot, males were injected with either saline or nicotine immediately prior to being placed in a novel environment for 40 min, during which locomotor activity was assessed by the number of times the animal interrupted a light beam during each minute. Each time point shows the number of beam crossings in that minute, shown as average plus/minus s.e.m. for all animals tested. Importantly, here and throughout the manuscript, the listed number of animals represent the number of litters analyzed, as we only assess one animal per litter in a given assay. Data are shown for saline injection ('Baseline') – exploratory behavior decreases over time in saline-injected animals as they habituate to the locomotor cage – and for 1.5 mg/kg nicotine injection in animals naïve to nicotine (Day 1) or following five or eight prior days of the same nicotine injection and locomotor assessment protocol.

The following figure supplements are available for figure 1:

**Figure supplement 1.** Physiological effects of nicotine exposure on treated males.

**Figure supplement 2.** Paternal nicotine exposure does not affect offspring anxiety-related behaviors.

**Figure supplement 3.** No significant effects of paternal nicotine exposure on offspring locomotor response to nicotine.

## Offspring of nicotine-treated males exhibit enhanced resistance to nicotine toxicity

Nonetheless, a clear phenotype emerged serendipitously from the SA paradigm. We found that in our strain background, the escalating nicotine dosing schedule of SA resulted in death of nearly all animals tested at the highest doses used. Surprisingly, NIC offspring survived for many more days, on average, than TA offspring (*Figure 2B*). This difference in survival was highly significant (Gehan-Breslow-Wilcoxon p<0.0001). As there was no difference in the daily levels of nicotine administered by either group (*Figure 2A*), this result suggests that paternal nicotine exposure can protect offspring from nicotine toxicity.

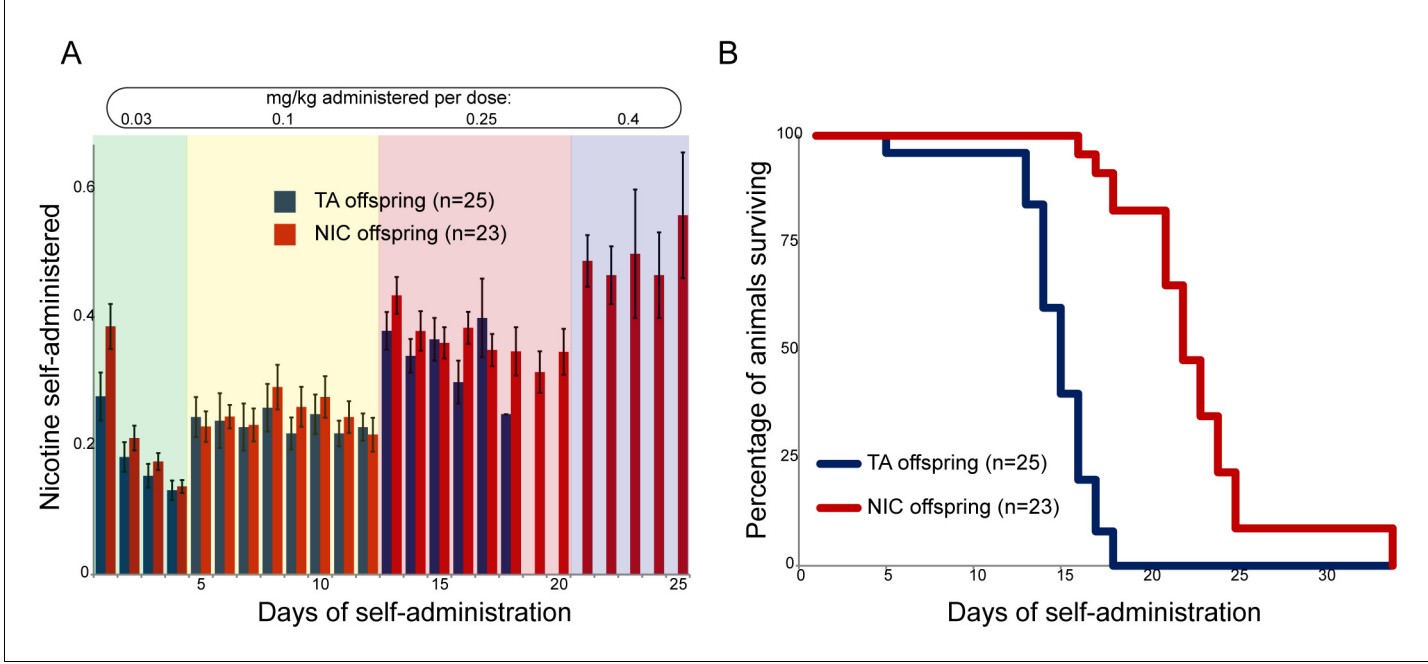

**Figure 2.** Paternal experience affects nicotine toxicity, but not self-administration, in offspring. (**A**) Paternal nicotine exposure does not affect nicotine self-administration in offspring. Each day, a mouse trained to self-administer nicotine (Materials and methods) was connected to the self-administration apparatus for one hour, with the dose of nicotine administered via cannula for every correct nose poke ramping up every 4–8 days, as indicated. Total nicotine self-administered is shown for each day of the protocol as average and s.e.m. Note that the numbers of animals participating in the trial decreased over time due to removal from the protocol (clogged catheter) or death – the listed n represents all animals that remained on the protocol until death. (**B**) Offspring of nicotine-exposed fathers exhibit significant protection from nicotine toxicity. Survival curve is shown for all animals on the self-administration protocol (underlying data are provided in *Figure 2—source data 1*). Nicotine offspring exhibited significantly increased survival during the time course of the assay relative to control offspring (Kaplan-Meyer survival curve, $p<0.0001$ for both Log-rank test and Gehan-Breslow-Wilcoxon test).

The following source data and figure supplement are available for figure 2:

**Source data 1.** Offspring of nicotine-exposed fathers exhibit significant protection from nicotine toxicity.

**Figure supplement 1.** Modest effect of paternal nicotine exposure on dietary training.

As TA and NIC offspring exhibit differences in their resistance to lethal doses of nicotine despite no difference in the daily level of nicotine consumed, we asked whether the effect of paternal nicotine exposure on offspring survival could be recapitulated using a single dose nicotine challenge, rather than the laborious self-administration protocol described above. This nicotine challenge was performed using two distinct paradigms. First, we simply challenged offspring of control or nicotine fathers with a single dose injection of nicotine – these 'naïve' animals had had no prior direct exposure to nicotine. In addition, we reasoned that since the animals in the self-administration paradigm were consuming nicotine for several weeks prior to eventual exposure to lethal levels of the drug (*Figure 2B*), this would be expected to substantially alter nicotine-related biology in the tested animal. We therefore also subjected TA and NIC offspring to one week of chronic low-dose nicotine (supplied in the drinking water) – we refer to these animals as the 'chronic' cohort – then challenged these animals with an injection of a single LD50 dose of nicotine.

As shown in *Figure 3A*, naïve TA and NIC offspring exhibited no significant difference in susceptibility to a toxic nicotine injection, indicating that paternal nicotine exposure does not program a constitutively nicotine-resistant state. In contrast, and consistent with the results of the self-administration test, male (but not female) offspring of nicotine-exposed fathers became significantly more tolerant to a lethal nicotine challenge than control offspring (*Figure 3B*), but only once they had become acclimated to a week of chronic nicotine. Taken together, these data demonstrate that

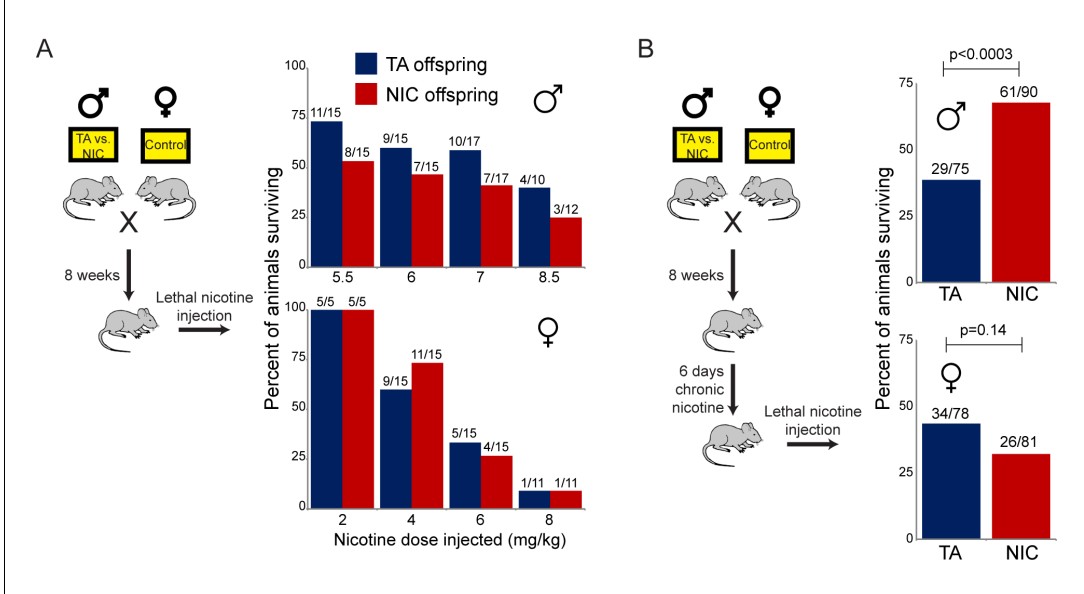

**Figure 3.** Paternally-induced protection from nicotine toxicity is primed by nicotine exposure in offspring. (**A**) Survival of TA or NIC offspring following a single injection of nicotine at the indicated dose. Above each bar, fraction shows the number of surviving animals over number of animals injected. For all four doses tested, there was no significant difference in toxicity between TA and NIC offspring (p>0.7 across all four doses for males, p>0.8 for females). (**B**) Survival of TA and NIC offspring following a single injection of nicotine at roughly the LD50 for naïve animals in (**A**) – 7.2 mg/kg for male offspring, shown in the top panel, 5.04 mg/kg for females, shown in the bottom panel. Here, offspring were acclimated to chronic nicotine in their drinking water for 6 days, with nicotine challenge being administered 24 hr following the last day of nicotine consumption.

male offspring of nicotine-exposed fathers exhibit an enhanced ability to develop tolerance to toxic doses of nicotine, but that this tolerance is only revealed following prior exposure to sub-lethal levels of nicotine.

## Paternal nicotine exposure affects xenobiotic clearance in offspring

What is the physiological basis for the enhanced resistance to nicotine toxicity observed in NIC offspring relative to TA offspring? Lethal doses of nicotine induce seizures originating in the hippocampus (*Fonck et al., 2003*). Resistance to such seizures could result from highly specific resistance mechanisms such as downregulation of nicotinic acetylcholine receptors in the hippocampus, or from relatively nonspecific resistance mechanisms such as enhanced detoxification of xenobiotics in the liver. Although we cannot definitively rule out a neural basis for the enhanced nicotine resistance observed in NIC offspring, several lines of evidence – including extensive RNA-Seq analysis of isolated hippocampus – argue against this resistance resulting from altered neural physiology (*Figure 4—figure supplement 1*, *Supplementary file 1*).

In contrast to the lack of relevant molecular changes observed in the brains of NIC offspring, we discovered a significant effect of paternal nicotine exposure on hepatic detoxification of nicotine in offspring. As shown in *Figure 4A*, nicotine-acclimated NIC offspring exhibit significantly higher levels of the long-lived nicotine metabolite cotinine at earlier time points after nicotine injection than do TA offspring. This finding is consistent with enhanced nicotine clearance underlying the nicotine resistance phenotype displayed by these animals, suggesting that paternal nicotine exposure programs a state of enhanced metabolic tolerance in offspring.

What is the molecular basis for the enhanced nicotine detoxification observed in NIC offspring? As the liver is the primary site of nicotine and other xenobiotic clearance in mammals, we investigated changes in mRNA abundance in hepatocytes isolated from TA and NIC offspring (*Figure 4B–C*, *Supplementary file 2*). Paternal nicotine exposure significantly (adjusted p<0.05) affected the expression levels of 51 genes, with upregulated genes being significantly enriched for those involved in lipid metabolism (p=3.9e-14), amino acid catabolism (p=6.6e-8), and various mitochondrial annotations including mitochondrial membrane (p=1.9e-7) (*Figure 4D–E*, *Figure 4—figure supplement*

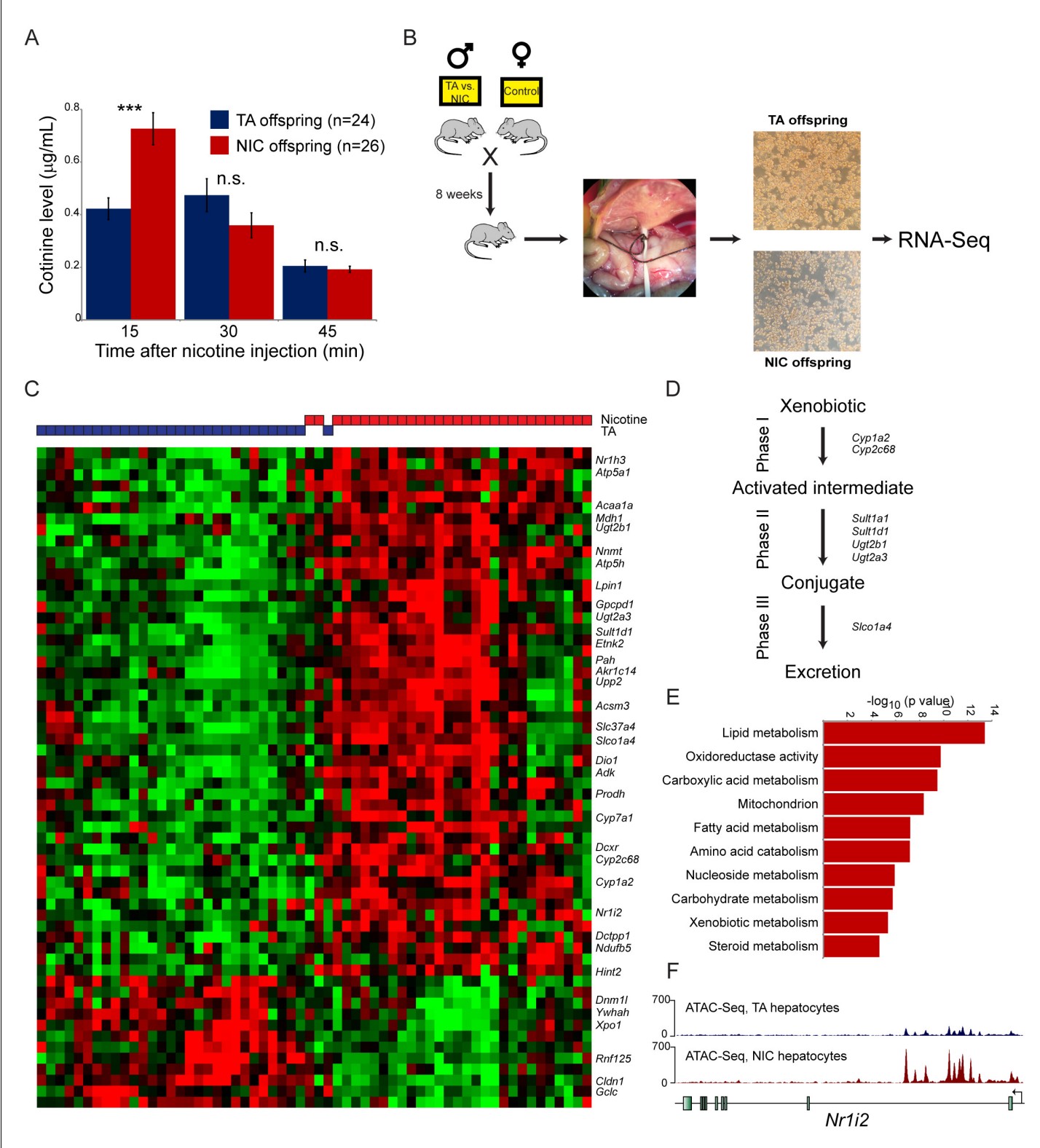

**Figure 4.** Paternal nicotine exposure induces an exaggerated protective response to xenobiotics. (**A**) Paternal nicotine exposure enhances nicotine metabolism in offspring. Male TA and NIC offspring were acclimated to nicotine for 6 days, then 24 hr later were injected with 1.5 mg/kg nicotine. Serum levels of the long-lived nicotine metabolite cotinine were measured at the indicated times after nicotine injection, with significantly (p<0.0002, t-test with Holm-Sidak correction) elevated cotinine levels being observed at the earliest time point analyzed, indicating enhanced nicotine clearance in NIC offspring. (**B**) Schematic of hepatocyte RNA-Seq experiment. (**C**) Cluster of hepatocyte RNA-Seq dataset. For each paternal treatment group (TA or

*Figure 4 continued on next page*

*Figure 4 continued*

NIC), data are shown for ten individual male offspring from ten separate litters, with hepatocytes from five animals also being cultured for varying times (0 to 21 hr) following isolation. Data are z score normalized for each culture time point. The heatmap shows 60 genes (filtered for average expression >25 ppm) changing with a multiple hypothesis-corrected p value<0.1. Underlying data are provided in *Figure 4—source data 1*. (D) Genes upregulated in NIC offspring encode enzymes involved in all three phases of xenobiotic metabolism, as indicated. (E) Selected Gene Ontology categories enriched among genes upregulated (adjusted p<0.1) in NIC hepatocytes. (F) ATAC-Seq coverage for TA and NIC hepatocytes, as indicated, across *Nr1i3*. See also *Figure 4—figure supplement 3*.

The following source data and figure supplements are available for figure 4:

**Source data 1.** Cluster of hepatocyte RNA-Seq dataset.
**Figure supplement 1.** Paternal nicotine has no significant effects on offspring hippocampal gene regulation or neural activity.
**Figure supplement 2.** Paternal nicotine exposure affects multiple phenotypes in offspring.
**Figure supplement 3.** Global differences in hepatocyte chromatin architecture between TA and NIC offspring.

*2A–B*). Most notably, given the nicotine resistance observed at the organismal level, NIC hepatocytes also exhibited increased expression of genes involved in drug metabolism (p=4.3e-6), with upregulated genes including 'Phase I' (*Cyp1a2*, *Cyp2c68*) and 'Phase II' (*Ugt2a3*, *Ugt2b1*, *Sult1d1*, and *Sult1a1*) detoxification enzymes, 'Phase III' membrane transporters (*Slco1a4*), as well as genes encoding the xenobiotic-responsive nuclear hormone receptors CAR and PXR (*Nr1h3* and *Nr1i2*) (*Figure 4C–D*). In addition, the primary cytochrome involved in nicotine clearance in rodents, *Cyp2a5*, was upregulated ~2 fold on average in NIC hepatocytes. Although this upregulation was not significant (adjusted p=0.2) in the genome-wide dataset due to sample to sample variability in expression of this gene, we validated upregulation of *Cyp2a5* in additional intact livers (n = 6 NIC, n = 4 TA, p<0.01) by q-RT-PCR (*Figure 4—figure supplement 2C*).

These gene expression studies thus reveal that, relative to TA hepatocytes, NIC hepatocytes exhibit a general derepression of target genes for a broad range of nuclear hormone receptors. To investigate the mechanistic basis for this derepression, we characterized open chromatin genome-wide in TA and NIC hepatocytes (n = 8 samples each) using ATAC-Seq (*Buenrostro et al., 2015*). Our ATAC-Seq dataset exhibited expected features such as strong peaks of accessibility over promoters and other regulatory elements (*Figure 4—figure supplement 3*). Comparing TA and NIC datasets, we observed a consistent global difference in overall chromatin accessibility – normalized ATAC peaks at regulatory elements were nearly 2-fold higher in NIC hepatocytes than in TA hepatocytes, while TA hepatocytes exhibited a consistently higher background of transposition throughout regions of the genome distant from regulatory elements (*Figure 4—figure supplement 3A–C*). Whatever the basis for this global change in chromatin accessibility, we additionally identified 1861 peaks of chromatin accessibility (*Figure 4F*, *Figure 4—figure supplement 3D–H*, *Supplementary file 3*) that differ significantly between TA and NIC hepatocytes after correcting for the global difference in peak height between these samples. Consistent with the changes in mRNA abundance observed in hepatocytes, these peaks were significantly enriched near genes involved in lipid metabolism (p=2.8e-18) and xenobiotic metabolism (p=1.3e-6), along with many related GO categories. We conclude that a history of paternal drug exposure can influence the chromatin landscape of hepatocytes in offspring, resulting in a broad increase in accessibility at regulatory elements involved in metabolism and detoxification.

## Enhanced xenobiotic resistance in NIC offspring is not specific for nicotine

Importantly, the gene expression program observed in isolated hepatocytes includes a broad variety of genes associated with drug metabolism, most of which are not specific for nicotine clearance. To test the hypothesis that the nicotine-resistant state of NIC offspring reflects a general xenobiotic response, rather than a nicotine-specific detoxification pathway, we asked whether NIC offspring also exhibit enhanced resistance to another toxic challenge, cocaine. As cocaine and nicotine

operate through distinct molecular pathways – cocaine prevents dopamine reuptake at the synaptic cleft by binding to and blocking the dopamine transporter, while nicotine activates and desensitizes nicotinic acetylcholine receptors – a finding of enhanced tolerance to cocaine would strongly argue against NIC offspring exhibiting specific epigenetic effects on the direct molecular receptor for nicotine.

We first assessed cocaine toxicity in 'naïve' animals that had not been previously directly exposed to nicotine or cocaine. Similar to our findings with nicotine toxicity (*Figure 3A*), naïve NIC and TA animals did not exhibit significant differences in their resistance to cocaine toxicity (*Figure 5A*). However, as the enhanced ability of NIC offspring to survive toxic nicotine levels was only revealed following pre-exposure of these animals to sub-lethal doses of nicotine (*Figure 3B*), we next sought to determine whether acclimation of NIC offspring to cocaine could induce a cocaine-resistant state. To address this question, TA and NIC offspring were chronically treated with sub-lethal doses of

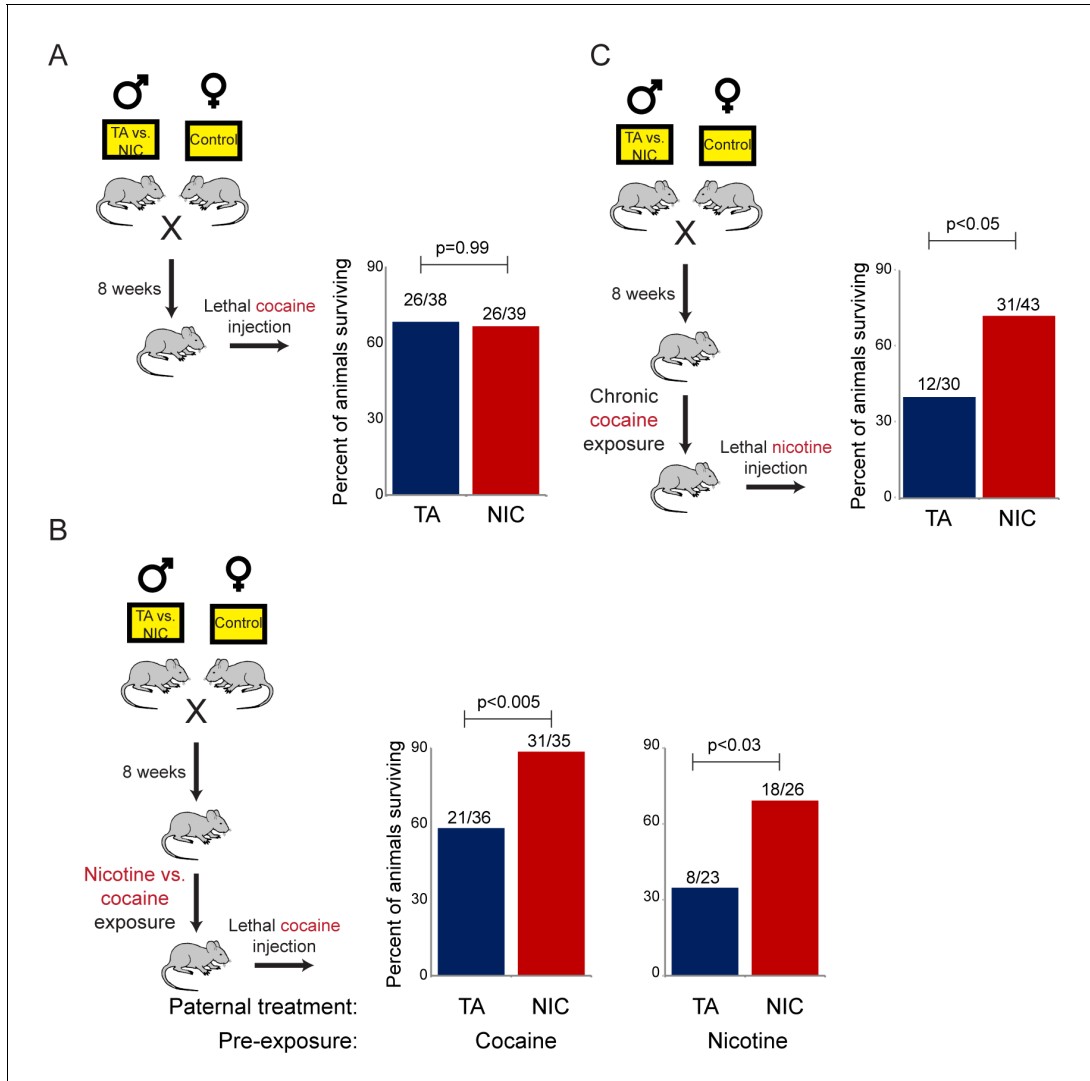

**Figure 5.** NIC offspring are protected from multiple xenobiotics. (**A**) Paternal nicotine exposure does not affect susceptibility of drug-naïve offspring to cocaine toxicity. Male TA and NIC offspring were injected with a single 100 mg/kg dose of cocaine. Survival is shown as in *Figure 3*. (**B**) Acclimation of TA and NIC offspring to either nicotine or to cocaine reveals protective effect of paternal nicotine exposure on offspring cocaine resistance. As in (**A**), for male offspring acclimated to chronic nicotine (200 µg/mL nicotine free-base in drinking water for six days) or cocaine (twice-daily injections with 15 mg/kg cocaine for five days). Twenty-four hours following final drug exposure, animals were injected with a single 100 mg/kg dose of cocaine. (**C**) Cocaine acclimation induces nicotine resistance in NIC offspring. Here, male TA and NIC offspring were acclimated to cocaine injections (twice-daily, 15 mg/kg) over five days. Twenty-four hours after the final cocaine injection, animals were injected with 7.2 mg/kg nicotine.

cocaine – twice-daily injections of 15 mg/kg cocaine for five days – prior to challenge with a toxic dose of cocaine. Astonishingly, this acclimation protocol resulted in enhanced resistance to cocaine toxicity in NIC offspring, relative to TA controls (*Figure 5B*), revealing that NIC offspring are hyper-responsive to multiple xenobiotics.

We next asked whether the process of acclimation to sub-lethal doses of nicotine or cocaine induces a drug-specific resistant state in NIC offspring. In other words, does pre-acclimation of NIC offspring to different molecules induce resistance specifically to the drug to which the animals were exposed, or do chronic exposures to multiple distinct drugs all induce a common state of general xenobiotic resistance? To distinguish these possibilities, we pre-acclimated TA and NIC offspring to either nicotine or cocaine, then challenged acclimated animals with a lethal dose of the drug to which they had not yet been exposed. Consistent with the hypothesis that drug acclimation induces a general xenobiotic response, we found that pre-acclimation to nicotine induced a cocaine-resistant phenotype in NIC offspring, and, conversely, that chronic cocaine could induce nicotine resistance (*Figure 5B–C*). Together, these data suggest that paternal nicotine exposure programs a hyper-responsive state in male offspring in which chronic xenobiotic exposure results in a generalized toxin resistance.

## Drug resistance is induced by multiple paternal drug exposures

The revelation that nicotine resistance in NIC offspring reflects a somewhat generic xenobiotic resistance program (*Figures 4C–D* and *5*) raises the question of what aspect of the paternal nicotine exposure paradigm is responsible for programming the offspring phenotype. The nicotine exposure paradigm utilized here induces nicotinic acetylcholine receptor (nAChR) signaling, with several physiological consequences: (1) nicotine dependence, (2) reduced caloric intake, and (3) physiological withdrawal resulting from the removal of nicotine for the final week prior to mating. To investigate the role of nAChR signaling in the paternal induction of offspring drug resistance, we made use of mecamylamine, a non-selective, non-competitive antagonist of nAChRs that readily crosses the blood-brain barrier.

Male mice were provided with 2.0 mg/kg/day mecamylamine via a surgically-implanted infusion pump, and mecamylamine-treated mice were split to either nicotine or TA drinking water, as in our primary nicotine exposure paradigm. Studies have previously shown that mecamylamine administration prevents known physiological responses to nicotine such as nicotine-induced anorexia (*Mineur et al., 2011*), hypothermia and locomotor effects (*Tapper et al., 2004*), and nicotine reinforcement (*Corrigall and Coen, 1989*). Male offspring of these fathers were then acclimated to nicotine for 6 days, then subject to a toxic nicotine challenge, as in *Figures 3* and *5*. Surprisingly, male mice concurrently treated with nicotine and its antagonist fathered offspring with the same enhanced nicotine resistance seen in NIC offspring (*Figure 6*). Importantly, this finding rigorously rules out the possibility that our nicotine exposure paradigm induces paternal effects on offspring as a consequence of the nicotine withdrawal stress imposed in the week before mating.

Moreover, the drug resistance observed in nicotine+mecamylamine offspring strongly argues that this paternal effect does not even require nicotine signaling in treated fathers, instead suggesting that the paternal effect is perhaps induced simply by exposure to xenobiotics. Consistent with this hypothesis, mecamylamine exposure alone also induced drug resistance in the next generation, although this effect was not as robust as that induced by nicotine or nicotine+mecamylamine (*Figure 6*). Together, these data demonstrate that drug resistance in sons can be induced by paternal exposure to both nAChR agonists and nAChR antagonists, arguing that paternal xenobiotic exposure is likely to be the relevant feature of our nicotine exposure paradigm.

## Relative sparing of hepatocytes following drug treatments in NIC offspring

Finally, we sought to understand the requirement for drug acclimation in revealing organismal drug resistance in NIC offspring. Curiously, the relative upregulation of xenobiotic processing genes (XPGs) in NIC offspring was observed in hepatocytes and livers isolated from 'naïve' animals that had not been exposed to nicotine or cocaine (*Figure 4*), yet enhanced resistance to toxins was only observed in animals that were first acclimated to one of these drugs (*Figures 3* and *5*). To test the hypothesis that XPG upregulation might be even stronger in NIC hepatocytes following drug

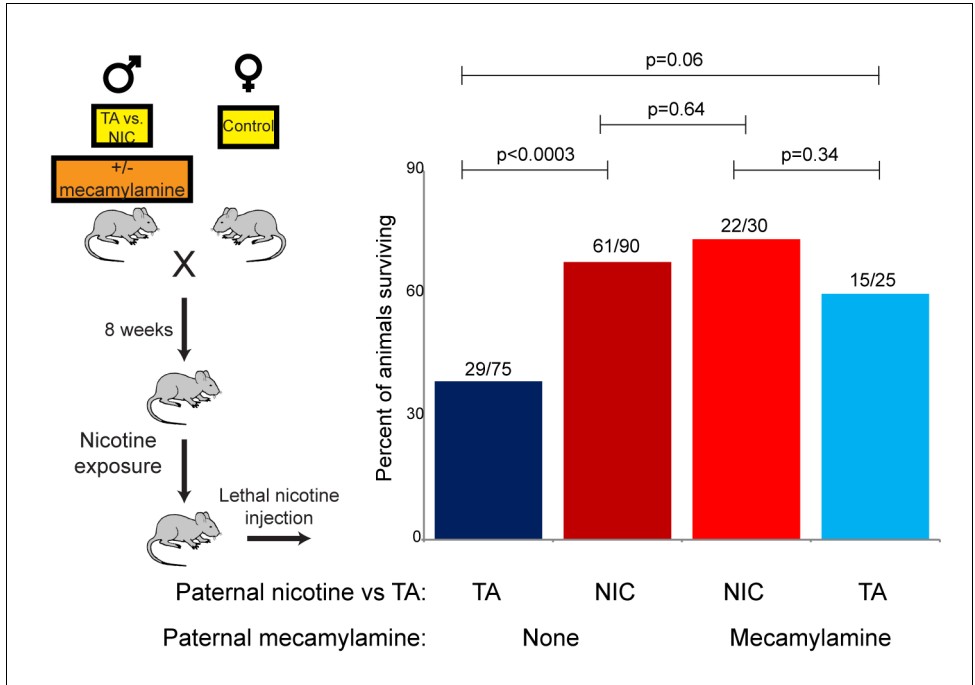

**Figure 6.** Offspring drug resistance is induced by a nicotine antagonist. Here, we modified the paternal exposure paradigm by implanting pumps to deliver the nicotine antagonist mecamylamine to male mice. Mecamylamine-treated mice were provided with nicotine or control solution for four weeks, then mated to control females. Male offspring were acclimated to chronic nicotine for six days and then subject to a toxic nicotine challenge, and survival is shown as in *Figures 3* and *5*. Data for no mecamylamine animals are reproduced from *Figure 3B*. Note that concurrent mecamylamine and nicotine exposure resulted in a protective effect on offspring, and even mecamylamine alone was able to modestly induce nicotine resistance in the next generation.

exposure, we set out to characterize gene expression changes in nicotine- or cocaine-acclimated offspring. However, in attempting to isolate hepatocytes from drug-acclimated TA and NIC offspring for RNA-Seq analysis, we noticed much poorer recovery of hepatocytes from TA than from NIC offspring (not shown), suggesting the possibility that NIC animals might be protected from drug-induced hepatotoxicity. Therefore, to quantify cell viability in vivo, we took a histochemical approach to assess apoptosis in livers from drug-acclimated TA and NIC offspring. Consistent with the relatively poor recovery of hepatocytes from TA animals, we observed substantial hepatocyte death in the livers of cocaine-exposed animals (*Figure 7A*). Importantly, while hepatocyte apoptosis and necrosis were extremely common in livers from cocaine-exposed TA offspring, NIC offspring were significantly protected from such cocaine toxicity (*Figure 7*). We conclude that the upregulation of XPGs in naïve NIC offspring is not sufficient to significantly protect animals from a lethal nicotine or cocaine challenge, but that this upregulation can protect hepatocytes from sub-lethal doses of these drugs. Following a week of chronic toxin exposure, TA offspring are left with substantially reduced liver function, while NIC offspring maintain greater numbers of functional hepatocytes. We speculate that this greater hepatocyte functional capacity, as well as the upregulation of XPGs in hepatocytes (*Figure 4*), may both serve to protect the animal from a single toxic dose of xenobiotic.

## Discussion

Here, we report a novel paradigm for intergenerational effects of paternal environment on offspring phenotype, based on paternal nicotine administration. Our data reveal that paternal nicotine exposure programs a state of nicotine resistance in offspring, but, surprisingly, neither the paternal sensing machinery nor the offspring response are specific for nicotine.

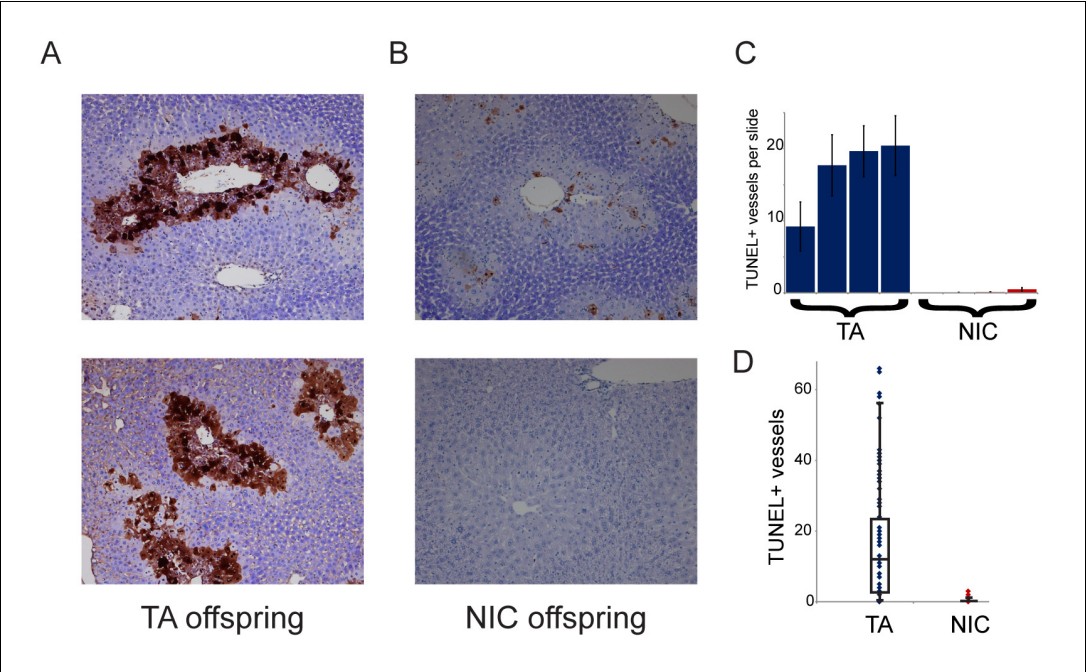

**Figure 7.** NIC offspring exhibit relative sparing of hepatocytes following chronic drug exposure. (A–B) Effects of chronic cocaine treatment on hepatocyte viability. Two representative sections are shown for TUNEL-stained livers from TA (A) and NIC (B) offspring following five days of cocaine injections (twice-daily, 15 mg/kg). Prominent centrilobular apoptosis is seen in TA offspring, but is almost completely absent in NIC offspring. (C–D) Quantitation of TUNEL staining data. (C) shows the average (plus/minus s.e.m.) number of TUNEL+ centrilobular regions per slide (staining of >25% of central vein circumference was counted as TUNEL+, and was assessed at five different levels for each liver lobe I-IV) for four individual TA (blue) and NIC (red) offspring, treated as in (A–B). (D) shows data for all individual slides as dots, with boxplot showing median, one standard deviation, and 5th/95th percentile for the 80 data points.

## Paternal nicotine exposure induces a pleiotropic, nonspecific set of phenotypes in offspring

The use of nicotine, a well-characterized small molecule that acts in vivo by binding to specific receptors, as the inciting paternal exposure enabled us to rigorously interrogate the specificity of the offspring response. Importantly, the enhanced toxin survival seen in offspring is not specific for the drug to which fathers were exposed – NIC offspring were hyper-resistant to both nicotine and to cocaine challenges – demonstrating that our paternal exposure paradigm does not result in transmission of a nicotine-specific phenotype to progeny (at least for toxicity, locomotor effects, and reward behavior). Mechanistically, the drug resistance observed in NIC offspring presumably results from the enhanced hepatic drug clearance observed in these animals (*Figure 4A*). Consistent with this increased nicotine clearance, isolated hepatocytes exhibited upregulation of a variety of xenobiotic processing genes (XPGs) accompanied by greater chromatin accessibility at relevant regulatory regions. A variety of XPGs are induced in NIC hepatocytes in addition to those known to play a role in nicotine clearance (*Figure 4C*), suggesting that NIC offspring may prove resistant to many toxins beyond the two tested in this study.

In addition to the significant derepression of xenobiotic response genes observed in NIC offspring, we note that the most significant effects of paternal nicotine on offspring hepatocyte gene expression occurred at metabolic genes (*Figure 4C,E*). This finding suggested that NIC offspring might also exhibit metabolic alterations, in addition to the documented changes in xenobiotic resistance. Alterations in glucose control and lipid metabolism are commonly observed in paternal effect studies, being observed not only in dietary paradigms but also in some stress and toxin-related paternal effect studies (*Rando and Simmons, 2015*), suggesting that multiple distinct stimuli

experienced by males might in some way convergently influence metabolic traits in offspring. As a detailed metabolic phenotyping of NIC offspring is beyond the scope of this study, we chose here to simply focus on the most common phenotype observed in other paternal effect experiments, assaying glucose and insulin tolerance in TA and NIC offspring (*Figure 4—figure supplement 2A–B*). Consistent with the ability of multiple paternal environments to alter glucose control in offspring, we observed that NIC offspring exhibited significantly diminished clearance of a glucose bolus, as well as a moderately diminished response to insulin.

Taken together, our data reveal (1) that paternal nicotine exposure induces a pleiotropic set of phenotypes in male offspring, and (2) that the induced phenotypes in offspring are not specific for nicotine. It will be of great interest in future studies to interrogate a wide variety of phenotypes in offspring of males subject to a broad range of exposure paradigms – including stress, nicotine treatment, and various diets – to identify common and divergent phenotypes induced by distinct paternal exposure paradigms.

## Paternal programming of offspring drug resistance is limited to male offspring

A curious feature of many, but not all, paternal effect paradigms reported in mammals is that phenotypic effects often manifest preferentially in offspring of one gender. For example, while paternal social defeat was reported to affect anxiety-related behavior in both male and female offspring, locomotor activity and sucrose preference were only altered in male offspring (*Dietz et al., 2011*). Here, we find that paternal nicotine exposure only affects drug resistance in male offspring, raising once again the unsolved question of why paternal environments induce gender-specific outcomes in progeny. Here, we consider three potential explanations for this phenomenon.

First, a subset of epigenetic information carriers – cytosine methylation and chromatin packaging – are associated in cis with a specific genomic locus, meaning that epigenetic changes occurring on the sex chromosomes will only affect progeny inheriting that chromosome. Thus, it is plausible that nicotine exposure affects epigenetic modification of the Y chromosome to program drug resistance in male offspring (or, less simply, that epigenetic marks on the X chromosome suppress an autosomal or small RNA-directed phenotype that would otherwise affect both male and female progeny). Second, X chromosome dosage compensation in mammals occurs via silencing of one of the two X chromosomes in females. The inactive X chromosome could thus act as a 'sink' for epigenetic silencing machinery in females (*Blewitt et al., 2005*), such that the effective levels of this machinery available for autosomal gene regulation could differ between males and females. In this scenario, paternal transmission of an epigenetically-marked autosomal locus, or RNA, could cause differential effects in developing male vs. female offspring based on differences in the available levels of epigenetic effector machinery. Finally, we note that an emerging theme in many paternal effect paradigms is that the phenotypic changes observed in offspring are known to be regulated by various nuclear hormone receptors (NHRs). For example, the phenotypes described in paternal stress paradigms are related to glucocorticoid receptor signaling, while the metabolic gene expression changes resulting from paternal dietary interventions exhibit significant overlap with genes regulated by NHRs such as PPARα (*Carone et al., 2010*). Here, we find that paternal nicotine exposure affects hepatic expression of many targets of metabolic NHRs, as well as the xenobiotic-responsive NHRs CAR and PXR (*Figure 4*). As sex hormones also act through NHR signaling – androgen receptor and estrogen receptor – we speculate that levels or activity of NHR coactivators or corepressors could differ in male vs. female progeny, resulting in altered penetrance or magnitude of paternal effects on NHR-mediated gene regulation.

## Offspring drug resistance is revealed by pre-exposure to xenobiotics

A crucial feature of the drug resistance exhibited by NIC offspring is that the toxin-resistant state is only revealed by pre-exposure of these animals to xenobiotics. This requirement for drug pre-exposure/acclimation emphasizes the key role of the *offspring*'s environment in the manifestation of an epigenetically 'reprogrammed' phenotype. In other words, the development of an animal's phenotype here involves an interaction between environmental conditions in two consecutive generations (see (*Rodgers et al., 2013*; *Zeybel et al., 2012*) for similar examples) – as with gene X environment effects, epigenetic marks also have context-dependent effects on organismal phenotype.

What is the mechanism by which low level drug exposure enhances the survival of NIC offspring? NIC hepatocytes exhibit derepression of xenobiotic response genes even before exposure to any drugs, yet these drug-naïve animals are no more resistant to nicotine or cocaine toxicity than control animals (*Figures 3A* and *5A*). Instead, the enhanced xenobiotic metabolism in NIC livers appears to protect susceptible hepatocytes from toxicity during a course of sublethal drug exposure (*Figure 7*). The loss of hepatocytes in drug-exposed TA animals presumably explains why fewer than 50% of these animals survive an LD50 dose – calculated using drug-naïve animals – of nicotine or cocaine (*Figures 3* and *5*), with the preservation of hepatic capacity in NIC offspring preventing this degradation in survivability. That said, not only do drug-acclimated NIC offspring simply *preserve* their survival in the face of an LD50 dose of these drugs, but they exhibit dramatically *improved* survival, as far more than half of these animals survive this challenge. We have yet to uncover the mechanistic basis for this enhanced survival, as RNA-Seq analysis of the hepatocytes isolated from drug-acclimated animals does not reveal further upregulation of XPGs than that documented for naïve hepatocytes (not shown). Future studies will investigate whether drug acclimation might (1) affect mRNA abundance in a limited subset of hepatocytes (which would be diluted out in whole liver or hepatocyte culture experiments), (2) affect mRNA abundance only transiently during drug exposure (and not in cultured hepatocytes), leaving behind higher levels of the encoded proteins without an mRNA-Seq signature, or (3) affect xenobiotic metabolism not at the level of mRNA abundance, but post-transcriptionally.

## How is nicotine sensed in exposed males?

The pleiotropic response observed in nicotine-exposed offspring raises the question of how nicotine is sensed in the paternal generation in this system. A key question in this regard is whether stress experienced by the nicotine-exposed males might be responsible for inducing the offspring phenotype, as it is known that a variety of paternal stress exposure paradigms – including early maternal separation, social defeat stress, and chronic variable low level stress – affect multiple phenotypes in offspring, from glucose control to anxiety-related behaviors (*Bale, 2015*). While we have not formally ruled out a role for paternal stress in our system – it will of course be of interest to assay offspring nicotine resistance in well-studied paternal stress paradigms – two findings strongly argue against this paternal effect arising from a general stress response. First, chronic exposure to the nicotinic receptor antagonist mecamylamine, which blocks nicotine dependence in nicotine-treated fathers, does not interfere with induction of xenobiotic resistance in offspring (*Figure 6*), thus definitively ruling out a role for paternal withdrawal stress in induction of this phenotype. This first point is further supported by the finding that mecamylamine alone – which on its own has little effect on anxiety, locomotor behavior, or physical withdrawal symptoms in nicotine-naïve mice (*Zhao-Shea et al., 2013*) – is sufficient to induce xenobiotic resistance in offspring. Second, in contrast to multiple reported paternal stress paradigms, we do not find any evidence that paternal nicotine exposure affects anxiety-related behavior in offspring (*Figure 1—figure supplement 2*).

What, then, is the relevant feature of nicotine in inducing xenobiotic resistance in offspring? Paternal effects on toxin resistance in offspring did not require nicotinic receptor signaling, as both nicotine itself as well as a nicotine antagonist were able to induce the protective response in offspring. As both nicotine and mecamylamine exposure can result in reduction of nAChR signaling via desensitization or antagonism, respectively, it is formally possible that nAChR deactivation is the inciting stimulus in the paternal generation (or, less likely, that the surgical stress of mecamylamine infusion pump implantation, and nicotine consumption, both convergently induce the same effect in offspring). However, we favor the simpler hypothesis that both of these molecules serve to program offspring drug resistance via effects on paternal xenobiotic sensing. This model naturally raises the question of how xenobiotic exposure is sensed. As a diverse variety of xenobiotics can affect gene regulation via activation of the NHRs CAR and PXR, these NHRs represent appealing candidates for the relevant xenobiotic sensor in fathers.

Whatever the nature of the relevant xenobiotic sensor, a key challenge to address is why experimental exposure to nicotine or mecamylamine (or, presumably, many other xenobiotics) reprograms offspring drug resistance relative to control animals, given that control animals are also exposed to a multitude of small molecules even in controlled laboratory conditions. Do nicotine and mecamylamine somehow induce a switch-like 'all or none' change in some epigenetic mark that is not present in control sperm, or is the overall activity level of a xenobiotic sensor translated into quantitative

changes in the levels of some continuous signal present in sperm? In the former case, what aspects of a given exposure paradigm are required to induce alterations to the sperm epigenome? We offer that one appealing mechanism for sensing increased levels of environmental xenobiotics would rely on comparing changes in sensor activity over an animal's lifetime. For instance, if CAR/PXR signaling early in life – in utero perhaps, or early in postnatal life – were to result in a long-lasting 'setpoint' for the levels of CAR/PXR activity expected later in life, then the organism could detect increased xenobiotic exposure later in life via changes in overall CAR/PXR activity compared to this setpoint. Future studies will explore the nature of the 'nicotine' sensor in the paternal generation, and how information about exposure history is transmitted to offspring.

Taken together, our studies define a novel paternal exposure paradigm based on a specific ligand-receptor interaction, and show that paternal nicotine exposure programs offspring for enhanced resistance to multiple distinct toxins. Our data also reveal broad metabolic gene expression changes in NIC offspring, with potential implications for metabolic and cardiovascular health of offspring. Future studies will determine whether paternal nicotine exposure affects offspring via epigenetic marks in the sperm (vs. seminal fluid, etc.), and how paternally-transmitted information alters the course of development to result in xenobiotic-resistant hepatocytes. It will also be of interest to extend these studies to human populations, where the longer half-life of nicotine could potentially result in self-administration phenotypes not observed in the mouse model.

## Materials and methods

### Animal husbandry and drug treatments

C57BL/6J mice (RRID:IMSR_JAX:000664), three weeks old, were obtained from Jackson labs on a weekly basis and group-housed (four mice/cage) on a 12 hr light-dark cycle (7:00 A.M. to 7:00 P.M). After arrival, males were immediately put on either tartaric acid (TA, 375 μg/ml) or nicotine (200 μg/ml nicotine free-base) in drinking water for five consecutive weeks, followed by an additional week on tap water prior to mating. Nicotine-exposed and control males were then allowed to mate (for six days) with control females from the same shipment date. F1 offspring from nicotine-exposed and control fathers were used for all experiments reported, generally at eight weeks of age unless otherwise noted. Animals were maintained on-site in accordance with an approved IACUC protocol (A-1788).

### Locomotor assay

F1 males from nicotine-exposed and control fathers were pre-conditioned to handling and injections with 0.9% saline (100 μl, i.p.) for three days prior to start of the study. For the nicotine test sessions, animals were injected with nicotine and transferred to individual cages placed within an infrared photobeam frame (San Diego Instruments). Test sessions lasted 40 or 90 min per day for nine consecutive days. Locomotor activity was defined as the number of beam breaks during a session, whereupon the animal had to cross at least two photobeams from the original location to count as ambulation. Results were statistically quantified using unpaired t-tests with multiple comparison adjustment (Holm-Sidak correction).

### Nicotine Self-Administration assay

Microsurgical catheter implant was performed on 7-week old F1 males from nicotine-exposed and control fathers. Animals were anaesthetized with ketamine (100 mg/kg BW) and xylazine (10 mg/kg BW) followed by a intrascapular and right midclavicular incision at the level of the carotid sheath. Blunt preparation was used to create a subcutaneous canal between the two incisions. Subsequently, the vena jugularis dextra was located and a catheter (2Fr, PV 10 cm, Instech Labs) was inserted and gently pushed forward into the vena cava superior, where it remained for the length of the study. The catheter was ligated to the vein using Ethibond Excel 4.0. The distal end of the catheter was connected to a button (25 G, VAB, Instech. Labs), which was placed subcutaneously in an intrascapular position for easy access. After verifying that there was no leakage, the incision sites were closed with Ethibond Excel 4.0. Through the catheter, the mouse was treated with heparin (15 I.U., Sigma-Aldrich, St. Louis, MO) and an antibiotic mix of Ticarcillin (20 mg booster, Sigma-Aldrich) and Amikacin (10 mg/kg BW, Sigma-Aldrich). Animals received Ketoprofen (5 mg/kg BW, Sigma-Aldrich) once

daily during a 3-day recovery phase. Afterwards, mice were put on a caloric restriction diet (85% w/w of regular 24 hr consumption) three days prior to start of the experiment. We preconditioned animals on sucrose pellets in a 60 min session once a day for seven consecutive days, whereby animals learned to nose-poke the active portal in a self-administration chamber in order to receive food reward. The number of nose-pokes required to get a sucrose pellet escalated starting with a fixed ratio of 1:1 (FR1) up to a fixed ratio of 5:1 (FR5). Only animals that had successfully been conditioned on sucrose pellets advanced to the testing phase, during which they administered nicotine to themselves through the implanted catheter. Catheter patency was verified daily by aspiration of blood and subsequent heparin infusion. Animals with blocked or dislocated catheters were excluded from the study. The self-administered nicotine doses started with 0.03 mg/kg/injection for 4 days, then 0.1 mg/kg/injection for 8 days, 0.25 mg/kg/injection for 8 days, and 0.4 mg/kg/injection for 8 days. The number of nose-pokes of the active versus the inactive portal, as well as the number of injections administered, were recorded and analyzed using GraphPad Prism 7.0 and multiple t-tests with Holm-Sidak correction. Survival was plotted as a Kaplan-Meier curve with significance levels calculated using modified Chi-square tests (Log-rank and Gehan-Breslow-Wilcoxon).

## Cotinine assay
Blood of F1 males from nicotine-exposed or control fathers was collected in EDTA-coated tubes after injection of 1.5 mg/kg nicotine free-base i.p. at 15 min, 30 min, and 45 min post-injection. Cellular components were separated from serum by centrifugation at 12,000 ×g for 10 min. Cotinine levels in serum of chronic F1s were measured using a Direct ELISA kit (CalBiotech Inc.). Samples were run as two technical replicates together with a cotinine standard curve for each 96-well plate. Analysis was performed using GraphPad Prism 7.

## Anxiety assays
The elevated plus maze consisted of four arms connected by a central axis (5 × 5 cm) and was elevated 45 cm above the floor. Two of the arms contained plastic black walls (5 × 30 × 15 cm) while the other two remained open (5 × 30 × 0.25 cm). Mice were individually placed on the center of the maze with their heads facing one of the open arms and allowed 5 min of free exploration. The number of entries into the open and closed arms, and the total time spent in the open and closed arms was measured by MED-PC IV software (MED associates, Inc.).The apparatus was thoroughly cleaned between animals. For activity in the open field, mice were placed in a rectangular arena made of Plexiglas (40 × 40 × 30 cm) and mouse activity was video recorded for 10 min. Total activity, velocity, and time spent in the peripheral and central area of the open field was analyzed using video tracking software (Noldus Ethovision).

## cFos staining and cell count
F1 males from TA- and nicotine-exposed fathers were treated as for transcriptome analysis and phenotype studies. Briefly, animals received nicotine in their drinking water (200 μg/ml nicotine free-base) for six consecutive days starting at seven weeks post-natum. Afterwards, mice were put on filtered tap water from 12:00 P.M. until 7:00 A.M. the next day followed by immediate tissue collection. Brains of additional eight-week old control animals are dissected 90 min after i.p. injection of 1.5 mg/kg BW nicotine free-base. Animals were anesthetized with sodium pentobarbital i.p. (200 mg/kg BW) followed by intracardial infusion of 10 ml ice-cold PBS and 10 ml paraformaldehyde (PFA; 4% w/v in PBS). Brains were kept at 4°C in 4% PFA for 2 hr and then transferred into 30% sucrose (w/v in PBS) until slice preparation.

Brains were sectioned using a microtome (Leica) into 25 μm slices and immersed in a 50% glycerol, 50% ethylene glycol solution (Sigma) to preserve the tissue. Brain slices were stored in −20°C until further processing. Using the free-floating immunostaining method, slices were washed with PBS for 5 min, permeabilized with 0.5% (v/v) Triton X-100 (Sigma) for 10 min, and blocked with 3% donkey serum for 30 min. The slices were incubated overnight at 4°C with antibodies against c-Fos (1:1000, catalog number: sc-52, lot number: D2315, Santa Cruz Biotechnology, Santa Cruz, CA). After washes with PBS, slices were incubated with Alexa Fluor 594 secondary antibodies (1:1000, ref number: A21207, lot number: 1602780, Life Technologies, Carlsbad, CA). Counterstaining was carried out with DAPI through mounting media (Cat number: H-1200, lot #: ZB0730, Vector, Burlington,

CA). Fluorescent images were captured using an AxioCam MRm camera (Carl Zeiss, Peabody, MA) attached to a Zeiss Axiovert inverted fluorescent microscope equipped with Zeiss filter sets 38HE, 49, and 20. Zeiss objectives A-p were subsequently processed using Axiovision version 4.8.2. Quantification of c-Fos-positive cells was performed using ImageJ, with a minimum of 6 hippocampal brain slices analyzed per animal.

## Tissue harvest for hippocampal mRNA-Seq

Seven week-old male F1 animals from control (TA) and nicotine-exposed fathers were divided into three treatment groups: naïve, chronic, and chronic + stimulation. Naïve mice were not exposed to nicotine before tissue collection at 8 weeks of age. Chronic animals received nicotine in their drinking water (200 µg/ml) for six consecutive days. Afterwards, chronic mice were put on filtered tap water from 12:00 P.M. until 7:00 A.M. the next day followed by tissue collection as for naïve animals. Chronic + stimulation animals were treated as chronic animals, but received an additional nicotine injection (1.5 mg/kg BW nicotine free base i.p.) 30 min before organ harvest. For all three sets of animals, following sacrifice brains were explanted and put on ice. A midline incision was executed and midbrain, hypothalamus, and hippocampus of either side were dissected. Tissues were immediately immersed in liquid nitrogen, then stored at −80°C until further processing.

## Hepatocyte isolation for mRNA-Seq and for ATAC-Seq

Eight week-old male F1 animals from control (TA) and nicotine-exposed fathers were anaesthetized using ketamine (100 mg/kg BW) and xylazine (10 mg/kg BW). The abdominal cavity was opened with a transverse incision below the rib cage. The portal vein was dissected with blunt forceps and a 26 G catheter needle was inserted. After cutting the vena cava inferior cranial of the liver, the organ was perfused firstly with 1X HBBS +200 mM EDTA (10 ml at 7 ml/min) and secondly with 50 ml DMEM containing collagenase type I (0.4 mg/ml) at 7 ml/min. The liver was then removed from the abdominal cavity, put in a petri dish containing culture medium (DMEM, 20% FBS, 1X ITS, 1X Penicillin/Streptomycin, 0.1 µM Dexamethasone, Sigma-Aldrich), and gently dissected to allow release of hepatocytes and supporting cells from connective tissue. Note that due to the disaggregation of the entire liver, mRNA abundance changes observed in a subset of hepatocytes (such as, for example, dying cells in drug-acclimated animals – *Figure 7*) will be diluted out by the majority of unaffected hepatocytes. After filtration through a 70 µm nylon cell strainer, cells were washed twice with PBS 1X and once with culture media (centrifugation at 500 rpm for 5 min), and plated on a 0.1% gelatin-coated well. Hepatocytes were allowed to adhere to the bottom of the well for three hours. Nonadherent cells were then removed, and fresh culture medium was then added, initiating our time course (T0, T1, T3, T21 hours). Cells were collected after a PBS 1X wash by adding TriZol to the well for RNA experiments.

## RNA-Seq

Strand-specific libraries were prepared as previously described (*Zhang et al., 2012*). Briefly, brain and liver were collected from nicotine-exposed and control F1 males. Hepatocytes were isolated as described above. For the hippocampus, after sectioning of brain into 1 mm slices, areas of interest were identified according to the Mouse Brain Atlas by Paxinos and Franklin and dissected using 0.5 mm punches.

RNA from brain and liver was isolated using standard TriZol protocols, followed by rRNA depletion (RiboZero kit, Illumina, Inc.). After first- and second-strand synthesis, adapters were ligated to fragments and amplified using multiplexed PCR primers. Libraries were sequenced on a NextSeq 500 platform from Illumina, Inc. Quality-controlled reads were aligned to the reference genome (Mus musculus/mm10) with Bowtie2 and differential expression was calculated using DESeq2. For multiple comparison adjustments, we used Holm-Bonferroni correction as a more conservative approach.

RNA-Seq data are available at GEO, accession # GSE94059.

## ATAC-Seq

ATAC-seq libraries were prepared for 16 hepatocyte samples (4 NIC and 4 TA animals, with each sample split into untreated and dexamethasone-treated aliquots) as previously described

(*Buenrostro et al., 2015*) using the Nextera DNA Library Preparation Kit (Illumina). Libraries were paired-end sequenced on a NextSeq 500, and reads were aligned to mm10 using Bowtie2, v2–2.1.0 with the parameters -D 15 R 2 N 1 L 20 -i S,1,0.50 –maxins 2000 –no-discordant –no-mixed. Mitochondrial DNA and random chromosome mapped reads were removed, and PCR duplicates were removed. Genome browser images were generated from merged datasets with reads extended to 150 bp, and normalized by total mapped reads per sample. For differential peak analysis, HOMER was used to identify NIC-specific peaks using TA peak files as background.

ATAC-Seq data are available at GEO, accession # GSE92240.

## Liver histology

Livers were harvested from F1 males from nicotine-exposed and control fathers under various conditions (pre-treatment with nicotine 1.5 mg/kg BW intraperitoneal b.i.d. for five days or cocaine 15 mg/kg BW intraperitoneal b.i.d. or acetaminophen 400 mg/kg BW q.d. for one day) and washed with PBS. A 4 mm slice was taken from each lobe and put in ice-cold 4% formaldehyde overnight. The next day, samples were dehydrated in a series of escalating ethanol solutions starting with 70% and ending with 100%, embedded in paraffin, and sectioned (4 μm slices), each section containing all four lobes, which were then mounted onto a glass slide. For H/E staining, slices were de-parafinized, incubated with xylene and a series of descending ethanol solutions. Incubation times for Mayer's hematoxylin (Sigma-Aldrich) and 1% Eosin Y (Sigma-Aldrich) were 30 s and 20 s, respectively. After dewaxing of tissue, TUNEL staining was performed following the manufacturer's recommendations (in Situ Cell Death Detection Kit, POD, Roche). Apoptotic areas per lobe were counted under a light microscope with 20X magnification at five different levels through the sample and analyzed with Image J.

## Acknowledgements

We thank members of the Rando, Tapper, and Gardner labs for discussions and critical reading of the manuscript. This work was supported by NIH grants F32DA034414 (MPV), R01DA033664 (OJR, ART, PDG), and R01HD080224 (OJR).

## Additional information

### Funding

| Funder | Grant reference number | Author |
| --- | --- | --- |
| National Institute on Drug Abuse | | Markus P Vallaster<br>Jennifer Ngolab<br>Rubing Zhao-Shea<br>Paul D Gardner<br>Andrew R Tapper<br>Oliver J Rando |
| Eunice Kennedy Shriver National Institute of Child Health and Human Development | | Shweta Kukreja<br>Xin Y Bing<br>Oliver J Rando |
| National Institutes of Health | F32DA034414 | Markus P Vallaster |
| National Institutes of Health | R01DA033664 | Paul D Gardner<br>Andrew R Tapper<br>Oliver J Rando |
| National Institutes of Health | R01HD080224 | Oliver J Rando |

The funders had no role in study design, data collection and interpretation, or the decision to submit the work for publication.

### Author contributions

MPV, Conceptualization, Formal analysis, Investigation, Methodology, Writing—original draft; SK, Validation, Investigation, Methodology; XYB, Formal analysis, Visualization; JN, RZ-S, Investigation; PDG, Conceptualization, Supervision, Funding acquisition; ART, Conceptualization, Resources,

Formal analysis, Supervision, Funding acquisition, Methodology, Writing—original draft, Project administration, Writing—review and editing; OJR, Conceptualization, Resources, Data curation, Formal analysis, Supervision, Funding acquisition, Writing—original draft, Project administration, Writing—review and editing

#### Author ORCIDs
Oliver J Rando, http://orcid.org/0000-0003-1516-9397

#### Ethics
Animal experimentation: This study was performed in strict accordance with the recommendations in the Guide for the Care and Use of Laboratory Animals of the National Institutes of Health. All of the animals were handled according to an approved institutional animal care and use committee (IACUC) protocol (A-1788) of the University of Massachusetts.

## Additional files

#### Supplementary files
• Supplementary file 1. RNA-Seq of TA and NIC hippocampus. RNA-Seq data for hippocampus isolated from male TA or NIC offspring, with samples being collected from drug-naïve animals, from 'chronic' animals subject to 6 days of nicotine administration prior to 24 hr of withdrawal, or subject to chronic nicotine exposure followed by a single sublethal injection of nicotine.

• Supplementary file 2. Hepatocyte RNA-Seq, naïve hepatocytes. RNA-Seq data for hepatocytes isolated from male TA or NIC offspring, then cultured in vitro for varying times as indicated.

• Supplementary file 3. Hepatocyte sATAC-Seq. List of peaks exhibiting significantly increased ATAC-Seq signal in NIC hepatocytes, relative to TA hepatocytes.

#### Major datasets
The following datasets were generated:

| Author(s) | Year | Dataset title | Dataset URL | Database, license, and accessibility information |
|---|---|---|---|---|
| Vallaster MP, Kukreja S, Rando OJ | 2017 | Hepatocyte RNA-Seq | https://www.ncbi.nlm.nih.gov/geo/query/acc.cgi?acc=GSE94059 | Publicly available at the NCBI Gene Expression Omnibus (accession no: GSE94059) |
| Rando OJ, Bing XY | 2017 | Hepatocyte ATAC-Seq | http://www.ncbi.nlm.nih.gov/geo/query/acc.cgi?acc=GSE92240 | Publicly available at the NCBI Gene Expression Omnibus (accession no: GSE92240) |

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
