## [Decision Letter]

Thank you for submitting your article "Intergenerational programming of hepatic xenobiotic response by paternal nicotine exposure" for consideration by *eLife*. Your article has been favorably evaluated by Detlef Weigel (Senior Editor) and three reviewers, one of whom, Kevin Struhl, is a member of our Board of Reviewing Editors.

The reviewers have discussed the reviews with one another and the Reviewing Editor has drafted this decision to help you prepare a revised submission.

Summary: Both reviewers agreed that the work is timely, and while the precise mechanisms are still unknown, the paradigm you have established likely presents a robust platform from which to investigate the fascinating phenomenon of intergenerational inheritance. As such, it is expected that the work will greatly help to advance the field of intergenerational inheritance.

Essential revisions: Both reviewers had access to the reviews from a previous journal that you provided with your submission, and your responses to the concerns raised in these previous reviews. These concerns especially addressed some of the technical aspects of the work, and the reviewers at *eLife* felt that your responses were not only adequate, but also that the additional analyses further elevated the work. Taking this into account, the reviewers agreed that the work should be published largely as is, with just a few clarifications:

What do you make of the fact that the transgenerational effects are only seen in male offspring?

In case you already have such data, it would be most interesting learn about the effects of other xenobiotics to see how broad the response is. Both reviewers agreed, however, that if such data are not in hand, they do not need to be added.

It is not entirely clear from the text alone what RNA-seq data were generated and compared (naïve and treated offspring). The Methods include a section on "Tissue harvest for hippocampal mRNA-Seq", but I could not find an equivalent section for the hepatocytes. In addition, this information should be in the main text and the legend to Figure 4. Also, in Figure 4, please indicate replicates from the same animal.

While it should be obvious to most readers, it would still be good to spell out why it is easier to confidently assess effects transmitted through the male rather than female germline.

More discussion of the surprising fact that both nicotine and its antagonist can elicit the same response would be helpful as well.

The exact comments of the reviewers are appended.

*Reviewer #1:*

An important question in modern biology is that of epigenetic information being passed on to later generations, often called transgenerational or intergenerational epigenetic inheritance. The difficulties are manyfold: First, it is often difficult to separate conventional maternal or paternal effects (due to direct transmission of informative RNA or protein molecules from parents to their progeny) from purely epigenetic changes linked to chemical modifications or protein decorations of DNA. Second, the phenotypes studied are usually complex ones, such as body weight, where it is often very difficult to pinpoint the underlying molecular causes. Finally, there is the question of how specific such transgenerational effects are, for example, whether exposure of the parents to a certain stress affects only the progeny's response to that or also to other stresses.

Vallaster and colleagues have established an excellent model to address many of these questions, using paternal exposure to nicotine, which acts through very well understood pharmacological and gene regulatory mechanisms. Exposure to nicotine was temporally separated from reproduction, to ensure that sperm or seminal fluid are not directly affected by nicotine. (The authors should either measure directly, or state how likely it is that there is measurable nicotine in sperm or seminal fluid during acute exposure. Rates of nicotine metabolism are probably known, and I would assume that there are references that could be cited.)

The authors found that the offspring were unchanged in their additive response to nicotine, but that they differed in protection from nicotine toxicity, which could be credibly traced back to rather specific changes in the expression of (and to a lesser extent, chromatin accessibility near) genes that enhance nicotine metabolism in the liver. While rather specific, this is not entirely specific, as the animals are also protected from cocaine toxicity.

A very nice experimental twist is that mecamylamine, a nonselective, noncompetitive nicotine antagonist has similar effects as nicotine, indicating that the response does not require nicotine signaling, but rather is related to a more generic xenobiotic response.

Although the work raises many interesting questions, I believe that there is great value in establishing this paradigm, which should be relatively easily replicable in other labs. I am looking forward to future work with this paradigm that almost certainly will more fully reveal how the xenobiotic treated offspring are primed, and what the tradeoffs of the priming are.

*Reviewer #2:*

This paper is concerned with the fascinating phenomenon of intergenerational programming. While this phenomenon has been clearly demonstrated in multiple ways, this paper is of particular interest and novelty. In particular, the authors use a paternal effect paradigm that involves a highly specific stimulus, nicotine, as opposed to more general metabolic stimuli, which permits them to analyze the specificity of the phenomenon. The most interesting observation is that the response in the subsequent generation is not specific for the original stimulus, but rather a more general xenobiotic response. This conclusion is validated by a considerable phenotyping. The authors also make the surprising observation that both nicotine and a nicotine antagonist of can induce this response. The paper also provides an initial mechanistic understanding in that the reprogammed mice increase nicotine clearance and hepatocytes show a transcriptional response linked to xenobiotic resistance. While the paper is somewhat limited in mechanistic understanding for how a specific stimulus results in a broader xenobiotic response, the phenomenon is important and there is enough mechanistic information to make the paper suitable for publication in *eLife*.

---

## [Author Response]

*Essential revisions: Both reviewers had access to the reviews from a previous journal that you provided with your submission, and your responses to the concerns raised in these previous reviews. These concerns especially addressed some of the technical aspects of the work, and the reviewers at eLife felt that your responses were not only adequate, but also that the additional analyses further elevated the work. Taking this into account, the reviewers agreed that the work should be published largely as is, with just a few clarifications:*

*What do you make of the fact that the transgenerational effects are only seen in male offspring?*

This is a great question – several other paternal effect paradigms (but not all) previously reported also show gender-specific phenotypes in offspring. We have now added a section to the Discussion providing some hypotheses for why these effects are gender-specific.

*In case you already have such data, it would be most interesting learn about the effects of other xenobiotics to see how broad the response is. Both reviewers agreed, however, that if such data are not in hand, they do not need to be added.*

We agree that this would be of great interest, but at present we have not looked into any additional xenobiotics or even initiated any such studies – given that we assay one animal per litter for drug resistance, each experiment of the type shown in Figure 3, Figure 5, or 6 requires at least 50 litters of animals.

*It is not entirely clear from the text alone what RNA-seq data were generated and compared (naïve and treated offspring). The Methods include a section on "Tissue harvest for hippocampal mRNA-Seq", but I could not find an equivalent section for the hepatocytes. In addition, this information should be in the main text and the legend to Figure 4. Also, in Figure 4, please indicate replicates from the same animal.*

We have edited the Methods to make this clearer, and we now indicate the replicates from a given batch of hepatocytes in the data table associated with Figure 4.

*While it should be obvious to most readers, it would still be good to spell out why it is easier to confidently assess effects transmitted through the male rather than female germline.*

We have added two sentences to this effect in the Introduction, although we honestly prefer the flow of the Introduction without this material.

*More discussion of the surprising fact that both nicotine and its antagonist can elicit the same response would be helpful as well.*

We have expanded on this part of the Discussion.

*The exact comments of the reviewers are appended.*

*Reviewer #1:*

*An important question in modern biology is that of epigenetic information being passed on to later generations, often called transgenerational or intergenerational epigenetic inheritance. The difficulties are manyfold: First, it is often difficult to separate conventional maternal or paternal effects (due to direct transmission of informative RNA or protein molecules from parents to their progeny) from purely epigenetic changes linked to chemical modifications or protein decorations of DNA. Second, the phenotypes studied are usually complex ones, such as body weight, where it is often very difficult to pinpoint the underlying molecular causes. Finally, there is the question of how specific such transgenerational effects are, for example, whether exposure of the parents to a certain stress affects only the progeny's response to that or also to other stresses.*

*Vallaster and colleagues have established an excellent model to address many of these questions, using paternal exposure to nicotine, which acts through very well understood pharmacological and gene regulatory mechanisms. Exposure to nicotine was temporally separated from reproduction, to ensure that sperm or seminal fluid are not directly affected by nicotine. (The authors should either measure directly, or state how likely it is that there is measurable nicotine in sperm or seminal fluid during acute exposure. Rates of nicotine metabolism are probably known, and I would assume that there are references that could be cited.)*

We have now provided nicotine and cotinine half-life information.